# Licochalcone D Induces ROS-Dependent Apoptosis in Gefitinib-Sensitive or Resistant Lung Cancer Cells by Targeting EGFR and MET

**DOI:** 10.3390/biom10020297

**Published:** 2020-02-13

**Authors:** Ha-Na Oh, Mee-Hyun Lee, Eunae Kim, Ah-Won Kwak, Goo Yoon, Seung-Sik Cho, Kangdong Liu, Jung-Il Chae, Jung-Hyun Shim

**Affiliations:** 1Department of Pharmacy, Mokpo National University, Jeonnam 58554, Korea; 17392303@mokpo.ac.kr (H.-N.O.); rhkrdkdnjs12@mokpo.ac.kr (A.-W.K.); gyoon@mokpo.ac.kr (G.Y.); sscho@mokpo.ac.kr (S.-S.C.); 2The China-US (Henan) Hormel Cancer Institute, Zhengzhou 450008, Henan, China; mhlee@hci-cn.org (M.-H.L.); kangdongliu@126.com (K.L.); 3Basic Medical College, Zhengzhou University, Zhengzhou 450001, Henan, China; 4College of Pharmacy, Chosun University, Gwangju 61452, Korea; eunaekim@chosun.ac.kr; 5Department of Dental Pharmacology, School of Dentistry, BK21 Plus, Jeonbuk National University, Jeonju 54896, Korea

**Keywords:** licochalcone D, non-small cell lung cancer, reactive oxygen species, apoptosis

## Abstract

Licochalcone D (LCD), a flavonoid isolated from a Chinese medicinal plant *Glycyrrhiza inflata*, has a variety of pharmacological activities. However, the anti-cancer effects of LCD on non-small cell lung cancer (NSCLC) have not been investigated yet. The amplification of *MET* (hepatocyte growth factor receptor) compensates for the inhibition of epidermal growth factor receptor (EGFR) activity due to tyrosine kinase inhibitor (TKI), leading to TKI resistance. Therefore, EGFR and MET can be attractive targets for lung cancer. We investigated the anti-proliferative and apoptotic effects of LCD in lung cancer cells HCC827 (gefitinib-sensitive) and HCC827GR (gefitinib-resistant) through 3-(4,5-dimethylthiazol-2-yl)-2,5-diphenyltetrazolium bromide (MTT) assay, pull-down/kinase assay, cell cycle analysis, Annexin-V/7-ADD staining, reactive oxygen species (ROS) assay, mitochondrial membrane potential (MMP) assay, multi-caspase assay, and Western blot analysis. The results showed that LCD inhibited phosphorylation and the kinase activity of EGFR and MET. In addition, the predicted pose of LCD was competitively located at the ATP binding site. LCD suppressed lung cancer cells growth by blocking cell cycle progression at the G2/M transition and inducing apoptosis. LCD also induced caspases activation and poly (ADP-ribose) polymerase (PARP) cleavage, thus displaying features of apoptotic signals. These results provide evidence that LCD has anti-tumor effects by inhibiting EGFR and MET activities and inducing ROS-dependent apoptosis in NSCLC, suggesting that LCD has the potential to treat lung cancer.

## 1. Introduction

Flavonoids are the most effective and variable biologically active compounds in plants. Licochalcone D (LCD) is an active flavonoid isolated from the Chinese medicinal herb *Glycyrrhiza inflata* [1]. LCD is present in the roots and rhizomes of *G. inflata*. The chemical name of LCD is (E)-3-(3,4-dihydroxy-2-methoxyphenyl)-1-[4-hydroxy-3-(3-methylbut-2-enyl)phenyl]prop-2-en-1-one. Various pharmacological actions, including anti-oxidant, anti-biotic, anti-ulcer, and anti-carcinogenic effects, have been described for LCD [2]. Previous studies have demonstrated that LCD can induce cell apoptosis and suppress migration and invasion in skin cancer [3]. In addition, LCD can suppress the proliferation in oral cancer cells [4]. However, the mechanism by which LCD exerts its effects on lung cancer has not been fully determined yet.

Non-small cell lung cancer (NSCLC) is the major cause of death from cancer in the world. NSCLC accounts for approximately 85% of lung cancer cases [5]. Chemotherapy remains marginally effective for NSCLC. Chemotherapy can slightly prolong the survival of patients with advanced lung cancer. However, it has clinically significant adverse effects [6]. The current treatment approach includes surgical resection, radiation therapy, and chemotherapy alone or in combination [6]. Despite these therapies, lung cancer is rarely curable, with an overall 5-year survival rate of only 15% [7]. As a result of the low cure rate of NSCLC, it is important to find effective treatment, with a focus on new molecular and targeted therapies.

Epidermal growth factor receptor (EGFR) mutations lead to the outstanding activation of EGFR signaling and carcinogenicity transformation both in vitro and in vivo [8]. Cancers with EGFR mutations (EGFR-mutated cancers) rely on EGFR signaling for growth. They are often sensitive to medical treatment with EGFR tyrosine kinase inhibitors (TKIs) [8,9]. Most patients with lung cancer have tumor-activating EGFR mutations. Treatment with EGFR-TKIs causes tumor reduction; however, the progression of cancer occurs at 6 to 12 months after treatment [10]. Various mechanisms of resistance to EGFR-TKIs (such as erlotinib and gefitinib) have been identified. The comprehension of these mechanisms is critical to evolving treatment strategies in the setting of resistance development. One of the resistance mechanisms is EGFR T790M point mutation within exon 20 [9]. hepatocyte growth factor receptor (MET) and human epidermal growth factor receptor (HER)2 overexpression with an upregulation of parallel signaling pathways have also been reported [9]. A gefitinib-resistant HCC827GR (*MET*-amplified) cell has been generated by exposing these cells to gefitinib for six months [11]. Their results showed that lung cancer cell growth was inhibited by simultaneous treatment with gefitinib and MET inhibitor. Thus, the dual inhibition of EGFR and MET might be a means to overcome lung cancer resistance.

The objective of this study was to investigate whether LCD could inhibit cell proliferation through EGFR and MET dual targets in NSCLC using human gefitinib-sensitive or resistant NSCLC cells. We found that LCD could induce the apoptosis of HCC827 and HCC827GR cells by inhibiting both EGFR and the MET signaling pathway. To confirm whether LCD induced apoptosis, we carried out cell proliferation, cell cycle distribution, reactive oxygen species (ROS) production, mitochondrial membrane potential (MMP) depolarization, and multi-caspase activation assays. The results of this study might shed light on the mechanism involved in the effect of LCD on lung cancer. We expect that LCD for cancer treatment might give improved results.

## 2. Materials and Methods

### 2.1. Reagents and Antibodies

LCD was prepared by Professor Goo Yoon according to previous reports [1]. Roswell Park Memorial Institute (RPMI)-1640 medium, phosphate-buffered saline (PBS), fetal bovine serum (FBS), penicillin/streptomycin, and trypsin were purchased from Hyclone (Logan, UT, USA). Dimethyl-sulfoxide (DMSO), 3-(4,5-dimethylthiazol-2-yl)-2,5-diphenyltetrazolium bromide (MTT), and Basal Medium Eagle (BME) were purchased from Sigma Chemical Company (St. Louis, MO, USA). Gefitinib was purchased from Cayman Chemical (Ann Abor, MI, USA). Savolitinib was obtained from Selleckchem (Houston, TX, USA). Primary antibodies against cyclin B1, cdc2, p21, p27, β-actin, Bid, Bcl-xl, Mcl-1, Bad, cytochrome c (cyto c), α-tubulin, COX4, apoptotic protease activating factor-1 (Apaf-1), cleaved poly (ADP-Ribose) polymerase (C-PARP), and ERBB3 (HER3) were obtained from Santa Cruz Biotechnology (Santa Cruz, CA, USA). Antibodies against phosphorylated (p)-EGFR (Tyr1068), EGFR, p-MET (Tyr1234/1235), MET, p-ERBB3 (Tyr1289), p-AKT (Ser473), and AKT were purchased from Cell Signaling Biotechnology (Beverly, MA, USA).

### 2.2. Cell Culture

The EGFR mutant (del E746_A750) NSCLC cell line HCC827 was obtained from the American Type Culture Collection (ATCC, Manassas, VA, USA). *MET*-amplified HCC827GR (gefitinib-resistant HCC827) cells were kindly contributed by professor Pasi A. Jänne (Department of Medical Oncology, Dana-Farber Cancer Institute, Boston, MA, USA). HCC827 and HCC827GR cells were cultured in RPMI-1640 medium supplemented with 10% FBS and 100 U/ml penicillin/streptomycin at 37 °C in a humidified atmosphere of 5% CO_2_.

### 2.3. Pull-Down Assay

To confirm the interaction between LCD and EGFR or MET, HCC827 and HCC827GR cell lysates were mixed with Sepharose 4B or LCD-Sepharose 4B beads. The protein extract was incubated with LCD-Sepharose 4B beads or Sepharose 4B beads in reaction buffer [50 mM Tris (pH 7.5), 5 mM EDTA, 150 mM NaCl, 1 mM/L dithiothreitol, 0.01% Nonidet P-40, 2 μg/mL bovine serum albumin, 0.02 mM phenylmethylsulfonyl fluoride, and 1X proteinase inhibitor] at 4 °C for 12 h. The mixture containing beads was washed six times with a washing buffer [50 mM Tris (pH 7.5), 5 mM EDTA, 150 mM NaCl, 1 mM dithiothreitol, 0.01% Nonidet P-40, and 0.02 mM phenylmethylsulfonyl fluoride]. Then, bound proteins were eluted with sodium dodecyl sulfate–polyacrylamide gel electrophoresis (SDS-PAGE) sample buffer and subjected to Western blot analysis.

### 2.4. Western Blotting

Whole cells were lysed with Radio-Immunoprecipitation Assay (RIPA) buffer (iNtRON Biotechnology, Korea). Protein concentrations were determined using a Bio-Rad DC Protein Assay kit (Bio-Rad, Hercules, CA, USA). An equal amount of protein was loaded on 8%–15% SDS-PAGE gels. After separation, proteins were transferred to polyvinylidene fluoride membranes (Millipore, Bedford, MA, USA), blocked with 3% or 5% skim milk in PBS containing 0.1% Tween-20 (PBST) at room temperature (RT) for 1 h or 2 h, and incubated with each primary antibody against its specific protein at 4 °C overnight. After washing six times, membranes were incubated with secondary antibodies for 2 h. Blots were scanned with an Image Quant LAS 500 (GE Healthcare, Uppsala, Sweden) using Western blotting luminol reagent (Santa Cruz, CA, USA).

### 2.5. ATP-Competitive Binding Assay

To determine whether LCD might compete with ATP, 100 ng of recombinant active EGFR or MET and ATP were pre-incubated at RT for 2 h. Subsequently, Sepharose 4B beads or LCD conjugated-Sepharose 4B beads were incubated at 4°C overnight and then washed with washing buffer. Bound proteins were subjected to Western blot analysis.

### 2.6. Kinase Assay

To determine the kinase activity of EGFR or MET in response to ATP, kinase reactions were incubated with EGFR (1.8 ng/μL; #3831), or MET (7 ng/μL; #3361), ATP (5 μΜ or 10 μΜ), substrates (0.2 μg/μL), LCD, gefitinib (1 μM), or savolitinib (5 nM) in a kinase reaction buffer containing 40 mM Tris (pH 7.5), 20 mM MgCl_2_, 0.1 mg/ml BSA, 50 μM dithiothreitol, 2 mM MnCl_2_, and 100 μM sodium vanadate using kinase enzyme systems (Promega, Madison, WI, USA). Before the plate was incubated at RT (22–25 °C) for 1 h, all kinase reactions were performed in 384-well plates with a volume of 5 μL. For the purpose of depleting the remaining ATP, 5 μL of ADP-Glo reagent (ADP-Glo kinase assay kit; Promega) was added to each well at RT for 40 min. Finally, 10 μL of kinase detection solution was added into each well of the 384-well plate. Luminescence reaction was proceeded and measured with a Centro LB 960 microplate luminometer (Berthold Technologies, Germany) for 0.5 s.

### 2.7. Molecular Modeling and Simulation

To predict the binding pose of the receptor tyrosine kinase, we performed a molecular docking simulation and a molecular dynamic (MD)s simulation. To perform the docking simulation, three-dimensional (3D) ligand and receptor structures were needed such as input files of the docking software, Vina. Receptor structures were downloaded from the protein database bank (PDB), including EGFR kinase in complex with gefitinib (PDB entry 2ITO) and MET kinase in complex with MK-2461 (PDB entry 3Q6W). LCD such as a ligand was made with Marvin sketch software. To run efficient searching, the assignment of the binding site was important for the docking simulation. Based on the binding interactions of the complex, the important ATP binding site was predicted to be Lys745 and Asp855 in EGFR and Lys1110 and Asp1222 in MET. Motifs of tyrosine kinase commonly included a glycine-rich nucleotide phosphate-binding loop, a G-loop (EGFR: Gly721-Gly724, MET: Gly1087-Gly1090), a hinge (EGFR: Leu792-Pro794, MET: Tyr1159-Lys1161), a Asp-Phe-Gly (DFG) motif (EGFR: Asp855-Gly857, MET: Asp1222-Gly1224), and an A-loop (EGFR: Asp855-Val876, MET: Asp1222-Leu1245). These binding poses were efficiently investigated for all important motifs. According to the result of the docking simulation, the top three binding poses were chosen based on the score of binding affinity. To confirm thermal stability in aqua solvent environment, MD simulation was performed using Gromacs software, which could overcome the lack of a rigid docking simulation. Complexes were solvated by a TIP3P water model and neutralized by adding ions. The Amber14SB force field (ff14SB) and general amber force field (GAFF) were applied for the protein and the ligand, respectively. Collected structures from the docking simulation were run in a physiological condition (310 K and 1 atm). After running for 500 ps in isothermal–isobaric (NPT) condition, canonical (NVT) ensemble was equilibrated for 30 ns. The stable structure was averaged in the converged state of MD simulation.

### 2.8. MTT Assay

HCC827 and HCC827GR cells were seeded into 96-well plates and incubated for 24 h. Then, these cells were treated with various concentrations of LCD for 24 h or 48 h. After incubation, MTT reagent was added to each well and incubated at 37 °C for 1 h. The culture medium and MTT were removed from each well, and the formazan in each well was dissolved with 100 μL of DMSO. The absorbance was measured at 570 nm using a Multiskan GO spectrophotometer (Thermo Scientific, Vantaa, Finland).

### 2.9. Anchorage-Independent Cell Growth Assay

Cells were seeded into 0.3% top agar over a layer of 0.6% bottom agar in a 6-well plate at a density of 8000 cells/well. Various concentrations of LCD and DMSO were added to the top and bottom layers containing culture medium (BME, 10% FBS, 2 mM L-glutamine and 5 μg/mL gentamicin). Plates were incubated at 37 °C for two weeks. The number of colonies was counted under a light microscope (Leica Microsystems, Wetzlar, Germany).

### 2.10. Annexin V/7-Minoactinomycin D (7-AAD) Staining

To evaluate NSCLC cell death with LCD treatment, Annexin V/7-AAD staining was performed using a Muse™ Annexin V and Dead Cell kit (MCH100105, Merck Millipore, Billerica, MA, USA). The HCC827 (1.95 *×* 10^5^) and HCC827GR (1.8 *×* 10^5^) cells were seeded onto a 6-well plate and treated with DMSO or LCD at different concentrations for 48 h. Cells were collected and subjected to Annexin V/7-AAD staining using 100 μL of Muse™ Annexin V and Dead Cell reagent according to the manufacturer’s protocol. Apoptotic cells were detected with a Muse™ Cell Analyzer (Merck Millipore).

### 2.11. Cell Cycle Analysis

A Muse™ Cell Cycle kit (MCH100106, Merck Millipore) was used to perform cell cycle analysis. Briefly, HCC827 and HCC827GR cells were collected by centrifugation at 4000 rpm for 5 min at 4 °C, washed three times with 1X PBS, and fixed with 70% cold ethanol at −20 °C for 24 h. These cells were collected by centrifugation at 4000 rpm for 10 min at 4 °C and washed once with PBS. Subsequently, Muse™ Cell Cycle Reagent was added to cell pellet followed by incubation at RT for 30 min in the dark. A Muse™ Cell Analyzer was used to obtain cell cycle data.

### 2.12. ROS Measurement

Intracellular ROS was measured with a Muse™ Oxidative Stress Kit (MCH100111, Merck Millipore). First, cells were grown in 6-well plates and treated with 5, 10, or 20 µM LCD for 48 h. Cells were washed with 1X assay buffer and incubated with a Muse™ Oxidative Stress Reagent working solution at 37 °C for 30 min. The level of fluorescence was determined with a Muse™ Cell Analyzer.

### 2.13. MMP Assay

MMP was measured using a Muse™ MitoPotential Kit (MCH100110, Merck Millipore). In brief, cells were exposed to 5, 10, or 20 µM of LCD for 48 h at 37 °C in a CO_2_ incubator. Cells were washed with 1× assay buffer, and fluorescence was then measured using Muse™ MitoPotential working solution. After incubation with 7-AAD for 5 min, the MMP was determined with a Muse™ Cell Analyzer.

### 2.14. Isolation of Cytosol and Mitochondrial Fractionation

Whole-cell extracts were obtained from LCD untreated or treated HCC827 and HCC827GR cells. Cells were resuspended in a plasma membrane extraction buffer containing 250 mM sucrose, 10 mM HEPES (pH 8.0), 10 mM KCl, 1.5 mM MgCl_2_∙6H_2_O, 1 mM EDTA, 1 mM EGTA, 0.1 mM phenylmethylsulfonyl fluoride, 0.01 mg/mL aprotinin, and 0.01 mg/mL leupeptin. Then, these cells were homogenized using 0.1% of digitonin and centrifuged at 13,000 rpm for 5 min. Supernatants were centrifuged at 13,000 rpm for 30 min to separate the cytosol fraction. The pellet was rinsed with plasma membrane extraction buffer and resuspended with 0.5 % of Triton X-100 in plasma membrane extraction buffer. Lysates were centrifuged at 13,000 rpm for 30 min to obtain supernatants as mitochondria fractions.

### 2.15. Multi-Caspase Assay

Multi-caspase (caspase-1, -3, -4, -5, -6, -7, -8, and -9) activity was analyzed with a Muse™ Multi-Caspase Kit (MCH100109, Merck Millipore). Briefly, HCC827 (1.95 *×* 10^5^ cells/well) and HCC827GR (1.8 *×* 10^5^ cells/well) cells were allowed to adhere for 24 h on 6-well plates. After treatment with LCD for 48 h, cells were harvested and washed with 1X caspase buffer. Then, these cells were incubated with Muse™ Multi-Caspase Reagent working solution at 37 °C for 30 min. After Muse™ Caspase 7-AAD working solution was added, flow cytometry analysis was carried out with a Muse™ Cell Analyzer.

### 2.16. Statistical Analysis

Statistical significance was evaluated using the software GraphPad Prism statistics (v5, GraphPad Software, USA, RRID: SCR_002798). Differences among multiple groups were tested using one-way or two-way ANOVA followed by Dunnett’s post hoc test. All data are expressed as mean ± standard deviation (SD). Differences were considered significant at *p* < 0.05.

## 3. Results

### 3.1. LCD Targets EGFR or MET

To understand the direct binding of LCD with EGFR or MET, we performed ex vivo pull-down assays (Sepharose 4B or LCD-Sepharose 4B beads) and in vitro ATP competitive binding assays. We used the gefitinib-sensitive NSCLC cell line HCC827 and gefitinib-resistant NSCLC cell line HCC827GR. As shown in Figure 1B, the pull-down assay and Western blotting results revealed that LCD bound to EGFR or MET in HCC827 and HCC827GR cells. However, there was no interaction between LCD and AKT. The results showed that the interaction between LCD and EGFR or MET was offset in the presence of 10 or 100 μM of ATP (Figure 1C). To further determine the interaction between LCD and EGFR or MET, we conducted kinase assay using gefitinib (EGFR inhibitor) and savolitinib (MET inhibitor) as positive controls. Compared to the untreated control group, EGFR and MET kinase activity were inhibited by treatment with LCD (Figure 1D). The inhibition levels of EGFR and MET kinase activity by LCD were similar to those by positive controls. These results suggest that LCD can suppress the kinase activity of EGFR or MET as an ATP competitive inhibitor. Figure 1E shows the predicted binding poses of LCD in EGFR and MET. In the complex of EGFR (Figure 1E left panel), LCD had two hydrogen bonds formed by the backbone of Met793 in the hinge and the sidechain of Asp855 in the DFG loop. The 4-hydroxy-3-(3-methylbut-2-enyl) phenyl group (A ring) and 3,4-dihydroxy-2-methoxyphenyl group (B ring) lay on the same plane and became jammed between the hydrophobic cores such as Leu718, Val726, and Ala743 of the P-loop and Leu844. In the complex of MET (Figure 1E right panel), the ketone group of LCD formed a hydrogen bond with the backbone of Met1160 in the hinge. The phenol ring (A ring) of LCD was surrounded by a hydrophobic pocket. Tyr1159 of the hinge and Ile1084, Val1092, Ala1108, and Lys1110 of the P-loop were covered similar to a lid. In addition, LCD was deeply supported by the hydrophobic sidechains of Met1160 of the hinge and Leu1140, Met1211, and Ala1221 of the ATP pocket downwards. The binding pose of EGFR was totally similar to that of MET driven by forming hydrogen bonds, and the hydrophobic interaction and exactly LCD was located at the ATP binding region of EGFR and MET. Notably, the stabilization of the complex would be enhanced by the hydrophobic interaction. Therefore, the predicted result was consistent with the experimental data, showing that LCD inhibited EGFR and MET competitively.

### 3.2. LCD Regulates Cellular Signaling Pathways Associated with EGFR or MET

To elucidate whether LCD could regulate EGFR and MET signaling cascades, Western blot analysis was used to evaluate the effects of LCD on expression and activation levels of proteins in RTK signaling pathways, including EGFR, MET, and their downstream signaling molecules. As displayed in Figure 2A, with an increasing concentration of LCD, the expression levels of p-EGFR, p-MET, and p-AKT showed a tendency to decrease in HCC827 and HCC827GR cells. These results indicate that LCD is involved in the down-regulation of the EGFR and MET signaling pathways.

### 3.3. LCD Inhibits the Growth of HCC827 and HCC827GR Cells

To analyze the effect of LCD on the viability of lung cancer cells, we used MTT to confirm the change in viability of HCC827 and HCC827GR cells. These two cell lines showed different gefitinib sensitivities. HCC827 cells were more sensitive to gefitinib than HCC827GR cells (Figure 3A). As shown in Figure 3A, LCD significantly decreased the viability of HCC827 cells. The same phenomenon was observed in HCC827GR cells (Figure 3A). The IC_50_ values of LCD for the viability of HCC827 and HCC827GR cells were 17.9 ± 0.97 μM and 19.1 ± 0.5 μM, respectively. To determine whether LCD could inhibit anchorage-independent growth, we carried out soft agar assays to measure the effects of LCD on HCC827 and HCC827GR cells. The number of colonies was decreased by treatment with LCD in a concentration-dependent manner (Figure 3B). These results show that LCD decreases the viability of NSCLC cell lines HCC827 and HCC827GR.

### 3.4. Cell Cycle Accumulation at G2/M Phase after Treatment with LCD

To assess the cause of the inhibited cell viability by LCD, we determined cell cycle distribution using a flow cytometric assay. The results indicated that the G2/M arrest of the cell cycle was increased in HCC827 cells and HCC827GR cells treated with LCD (5, 10, and 15 μM) for 48 h (Figure 4A). As shown in Figure 4B, the treatment of HCC827 and HCC827GR cells with LCD for 48 h resulted in a prominent increase in the expression of the Sub-G1 population in a dose-dependent manner. To confirm the underlying biochemical mechanisms related to the regulation of G2/M cell cycle arrest, we tested the effect of LCD on protein levels of cyclins and cyclin-dependent kinase (cdk)s during the G2/M cell cycle arrest progression. Treatment with LCD dramatically decreased the expression levels of cyclin B1 and cdc2 but increased p21 and p27 expression in HCC827 and HCC827GR cells (Figure 4C). The increased sub-G1 population of the cell cycle might suggest induced cell apoptosis. Thus, we performed flow cytometry analysis to confirm the level of apoptosis. HCC827 cells treated with LCD at concentrations of 5, 10, and 20 μM showed 16.1 ± 0.6%, 36.2 ± 0.3%, and 45.2 ± 0.6% of total apoptotic cells, respectively (Figure 4D). HCC827GR cells treated with LCD at the same concentrations of 5, 10, and 20 μM resulted in 12.7 ± 1.7%, 26.1 ± 0.7%, and 43.5 ± 0.4% of total apoptotic cells, respectively (Figure 4D). These results suggest that LCD can induce G2/M cell cycle arrest and apoptosis.

### 3.5. LCD Induces ROS-Dependent Apoptosis

To understand the mechanism of LCD-induced apoptosis, we tested the effects of LCD on intracellular ROS generation. Levels of ROS in HCC827 and HCC827GR cells were measured at 48 h after cells were treated with LCD. ROS values of HCC827 cells treated with LCD at concentrations of 5, 10, and 20 μM were 12.2 ± 1.2%, 21.5 ± 0.7%, and 34.3 ± 0.5%, respectively. ROS values of HCC827GR cells treated with LCD at concentrations of 5, 10, and 20 μM were 9.5 ± 0.3%, 18 ± 0.4%, and 31.3 ± 2.9%, respectively (Figure 5A). To further examine the significance of ROS in LCD-induced apoptosis, HCC827 and HCC827GR cells were pretreated with N-acetylcysteine (NAC), an ROS inhibitor, for 3 h and then treated with 20 μM of LCD for 48 h. NAC treatment resulted in similar ROS fluorescence levels compared to controls. However, ROS produced after treatment with LCD was markedly blocked by co-treatment with NAC (Figure 5B). As shown in Figure 5C, NAC significantly suppressed LCD-induced NSCLC cell growth inhibition. In addition, protein expression levels after co-treatment with LCD and NAC were determined by Western blot analysis using corresponding antibodies. Interestingly, the co-treatment of NAC and LCD did not restore the expression of p-EGFR and p-MET to control levels (Figure 5D). On the other hand, the expression of PARP was increased and the expression of C-PARP was decreased compared to LCD treatment (Figure 5D). These results indicate that LCD can induce ROS-dependent apoptosis in both NSCLC cells.

### 3.6. LCD Induces Apoptosis through the Mitochondrial Pathway

MMP is an important index to estimate the permeability of the mitochondrial membrane. To determine whether LCD can induce apoptosis by disrupting the mitochondrial membrane, NSCLC cells were stained with MitoPotential dye, a cationic and lipophilic dye, to monitor MMP. Results showed that the loss of MMP was increased drastically after treatment with LCD (Figure 6A). Western blot analysis was performed to check the levels of proteins associated with the mitochondria. As shown in Figure 6B, Bid was cleaved as the concentration of LCD increased. LCD increased the expression of a pro-apoptotic protein Bad (Figure 6B). In contrast, the expression levels of anti-apoptotic proteins Mcl-1 and Bcl-xl were decreased following LCD treatment (Figure 6B). During apoptotic cell death, early events that occur include mitochondrial depolarization and the loss of cyto c from the mitochondrial intermembrane space. Cyto c was increased in the cytosol fraction but decreased in the mitochondrial fraction of NSCLC cells after LCD treatment (Figure 6B). The expression of Apaf-1 known to form apoptosome with cyto c was also increased by LCD. Furthermore, the expression level of a DNA repair enzyme PARP was reduced by LCD in a dose-dependent manner (Figure 6B). To determine whether LCD-induced apoptosis was associated with caspase activation, levels of multi-caspase in NSCLC cells were assessed with a Muse™ Cell Analyzer. As shown in Figure 6C, LCD induced activities of multi-caspases dose-dependently (25%, 40%, and 46% for HCC827 cells and 28%, 35%, and 48% for HCC827GR cells by LCD at concentrations of 5, 10, and 20 μM, respectively). These results demonstrate that LCD sensitizes NSCLC cells to apoptosis via the mitochondrial pathway and the activation of caspases.

## 4. Discussion

Chemotherapy drugs generally include cisplatin and taxanes in the treatment of lung cancer. Unfortunately, many patients acquire resistance to the drug either intrinsically or after medication [6]. The occurrence of gefitinib resistance is a barrier to have effective clinical therapies for a number of solid tumors, including lung cancer. Various gefitinib resistance mechanisms, including EGFR mutations and *MET* amplification, have been reported in several studies [9,10]. More than 90% of EGFR mutations occur in exons 19–21, with exon 19 mutations being the most frequent [9]. The deletion of exon 19 and EGFR mutation of L858R of exon 21 have higher sensitivities than the TKI response in those with wild-type EGFR [12]. Conversely, resistance to TKI can be induced through the acquisition of secondary mutations of EGFR (T790M, L747S, D761Y, and T854A) or the activation of other signaling bypass pathways [9]. Resistance obtained after TKI treatment is almost unavoidable, and the success rate of treatment is low. Overcoming these resistances requires a strategy to target molecules related to resistance or in combination with other compounds.

Some clinical studies of EGFR- or MET-targeted inhibitors have been discontinued. However, others have shown encouraging results [10]. The overexpression of EGFR and MET has been reported in NSCLC, which can activate various downstream signaling molecules involved in cell growth and survival [8,13]. Blocking EGFR or MET alone for tumor suppression can lead to cell survival by activating other alternative pathways. Previous reports have shown that treatment with a single inhibitor of EGFR and MET, respectively, has no effect on cell proliferation, although cell survival is suppressed when EGFR and MET inhibitors are combined [11,14]. A dual blockade of EGFR and MET can significantly inhibit the proliferation several carcinoma cells, including lung cancer cells [14,15], head and neck cancer cells [16], and colon cancer cells [17]. Similarly, our data showed that treatment with EGFR and MET alone did not affect cell viability. However, they inhibited cell survival upon combination treatment. LCD inhibited lung cancer cell proliferation through ATP competitive inhibition of EGFR and MET as a single drug. Other studies have reported that inhibitors of EGFR and MET alone could inhibit cell survival, whereas a combination treatment of EGFR and MET inhibitors can significantly inhibit cell survival, showing synergistic effects on the induction of apoptosis [13].

In lung cancer cells, a signaling network exists between the same RTK family: EGFR, MET, and ERBB3. The results of the present study revealed that the levels of phosphorylated proteins of EGFR, MET, and ERBB3 decreased with an increasing concentration of LCD. Since the amplification of *MET* can induce gefitinib resistance by inducing ERBB3-dependent activation [11], it is thought that LCD targeting MET can also inhibit the expression of ERBB3. EGFR and MET can activate and share important downstream molecules involved in biological activities such as cell growth and survival [18]. The activation of AKT in lung cancer cells is involved in imparting resistance to TKI as well as cell growth and proliferation [10]. The reduction of p-AKT caused by the dual targeting of EGFR and MET might be a pathway to overcome resistance by preventing the conversion of compensation pathways. Thus, the dual targeting of EGFR and MET might be a promising therapeutic strategy to overcome lung cancer that is sensitive or resistant to TKI.

One of the meaningful findings in our study was that the mechanism underlying the apoptosis induced by LCD involved the inhibition of EGFR and MET. Apoptotic cell death is induced through two pathways: the extrinsic pathway and the intrinsic pathway. Apoptosis is caused by a variety of extrinsic and intrinsic factors such as ROS, DNA damage, and heat shock [19]. ROS can regulate physiological functions such as cell proliferation and cell cycle progression. However, excessive ROS levels can cause cell damage and lead to apoptotic cell death [19]. The exposure of NSCLC cells to LCD significantly increased ROS production, whereas ROS levels were restored to control levels by NAC. The expression of p-EGFR and p-MET, which were inhibited by LCD treatment, showed no significant difference by NAC. These results indicate that ROS is downstream of EGFR and MET, suggesting that ROS might be closely involved in the cell proliferation inhibition of LCD. ROS are known to affect EGFR signaling. ROS can induce TKI resistance by activating the EGFR signaling pathway [20,21]. High levels of ROS play an opposite role in tumor progression and drug resistance, leading to cell cycle arrest, apoptosis induction, and toxic effects on cancer cells [21]. This indicates that an appropriate level of ROS can mediate drug resistance, whereas excessive ROS production can lead to cell death. LCD can induce the apoptosis of lung cancer cells through the accumulation of ROS. ROS is closely associated with the mitochondrial pathway. In fact, the location of most intracellular ROS production is mitochondria [19]. LCD increased MMP loss and up- or down-regulated mitochondrial-related proteins, leading to apoptotic cell death through mitochondrial (intrinsic) pathways. Apoptosis molecular cascade by LCD led to the release of cyto c into the cytoplasm, the activation of Apaf-1 and caspase, and the cleavage of PARP. It is interesting to note that ROS plays an important role in the apoptosis-inducing mechanism of LCD because PARP, which was reduced by LCD, increased PARP expression by co-treatment with NAC.

ROS is also known to be involved in cell cycle progression, which is regulated by cyclin and cdk [20]. The link between cell cycle and apoptosis has been demonstrated in many studies [22]. It regulates cell proliferation. The cyclin B1/cdc2 complex is involved in the G2/M phase transition, and cdk activity is mediated by cdk inhibitors (CKI) such as p21 and p27 [22,23]. LCD decreased cyclin B1/cdc2 protein expression but increased CKI, causing G2/M cell cycle arrest. LCA and LCB of the same licorice family as LCD can also block G2/M cell cycle progression and induce the apoptosis of lung cancer cells [14,24]. Some agents that can induce apoptosis may also increase cytotoxicity in association with G2/M checkpoint arrest [23].

Importantly, LCD shows anti-tumor activity without causing weight loss in vivo using a xenograft model of oral cancer cells [4]. In addition, LCD (1 μg/ml) shows myocardial protective effects in the cardiac tissues of injured rats [25]. These findings indicate that LCD is low in toxicity. In addition, while common flavonoids generally have low bioavailability in humans, LCs have been shown to be well absorbed through passive diffusion through the Caco-2 cell monolayer, which is a human intestinal cell line [26]. Since intestinal permeability is an important factor influencing the bioavailability of drugs, LCs can be expected to have high bioavailability in the human body. Liquiritin, another component of licorice, enhanced the cell proliferation inhibitory effect of cisplatin in gastric cancer cells resistant to cisplatin, and showed synergistic effects on tumor growth inhibition in vivo [27]. In the future, we will evaluate the anti-tumor efficacy of lung cancer cells with different resistance mechanisms for the combination of anti-cancer drugs and LCD.

## 5. Conclusions

In summary, the results of this study suggested a molecular mechanism for the anti-cancer effect of LCD in gefitinib-sensitive or gefitinib-resistant lung cancer cells. The anti-tumor efficacy of LCD was shown through the dual inhibition of EGFR and MET activity, suggesting that it was a principal target of LCD. In addition, LCD induced ROS-dependent apoptotic cell death and inhibited the proliferation of lung cancer cells. In conclusion, LCD offers clinical benefit for TKI-sensitive or TKI-resistant lung cancer. It shows potential as an effective anti-cancer compound.

## Figures and Tables

**Figure 1 biomolecules-10-00297-f001:**
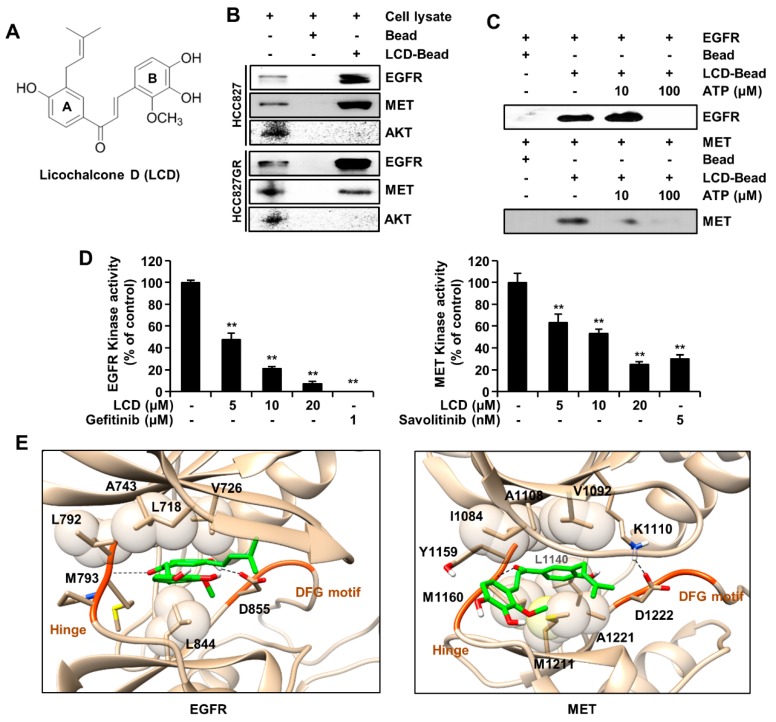
Licochalcone D (LCD) and epidermal growth factor receptor (EGFR) or hepatocyte growth factor receptor (MET) protein interaction. (**A**) Structure of LCD. (**B**) Pull-down assay. Whole cell lysates and Sepharose 4B or LCD-Sepharose 4B beads were incubated together. After washing, proteins bound to bead were released with SDS-PAGE loading buffer (lane 1, control; lane 2, Sepharose 4B beads; and lane 3, LCD-Sepharose 4B beads). (**C**) LCD binds to EGFR or MET competitively with ATP. Active EGFR (100 ng) or MET (100 ng) was incubated with ATP at different concentrations (0, 10, or 100 μM). Proteins were subjected to Western blotting. (**D**) EGFR or MET kinase activity of LCD by ADP-Glo kinase assay: percent inhibition rate shown at 5, 10, and 20 μM of LCD treatment. Data represent the mean value ± SD (n = 3). **, *p* < 0.01 compared with the control. (**E**) Predicted binding poses for EGFR (left) and MET (right). The dash line represented a hydrogen bond. LCD was tightly sandwiched between the hydrophobic sidechains (sphere). LCD was stabilized by hydrogen bonds and the hydrophobic interactions.

**Figure 2 biomolecules-10-00297-f002:**
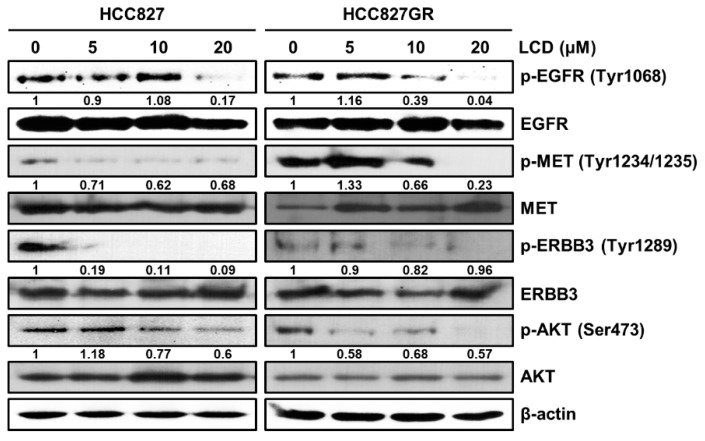
Comparison of EGFR or MET-related protein expression in HCC827 and HCC827GR cells. HCC827 and HCC827GR cells were treated with 5, 10, and 20 μM of LCD for 48 h. Levels of phosphorylated (p)-EGFR (Tyr1068), EGFR, p-MET (Tyr1234/1235), MET, p-ERBB3 (Tyr1289), ERBB3, p-AKT (Ser473), and AKT were analyzed by Western blotting. β-actin was used as a loading control. Band intensities were quantified using Image J.

**Figure 3 biomolecules-10-00297-f003:**
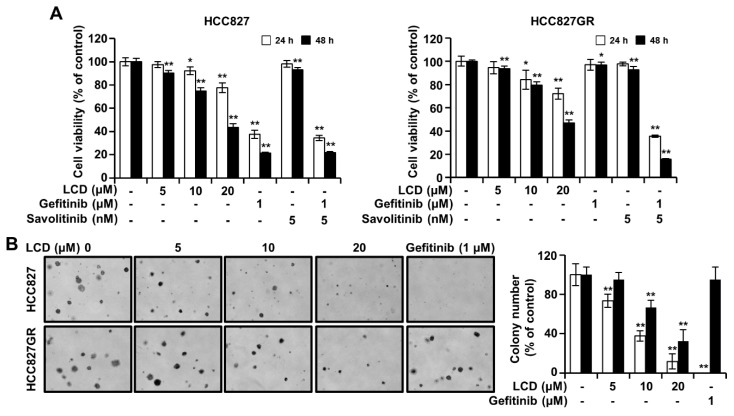
Effects of LCD on cell viability and colony formation. (**A**) HCC827 and HCC827GR cells were incubated in 96-well plates for 48 h in culture medium supplemented with LCD at an increasing concentration from 5 μM to 20 μM, and control cells. Cell viability was measured by 3-(4,5-dimethylthiazol-2-yl)-2,5-diphenyltetrazolium bromide (MTT) assay based on absorbance at 570 nm. Values shown in the graph represent mean ± SD (n = 6). (**B**) Anchorage-independent soft agar assay was performed with HCC827 and HCC827GR cell lines. Data are representative of triplicate experiments. *, *p* < 0.05 and **, *p* < 0.01 compared with the control.

**Figure 4 biomolecules-10-00297-f004:**
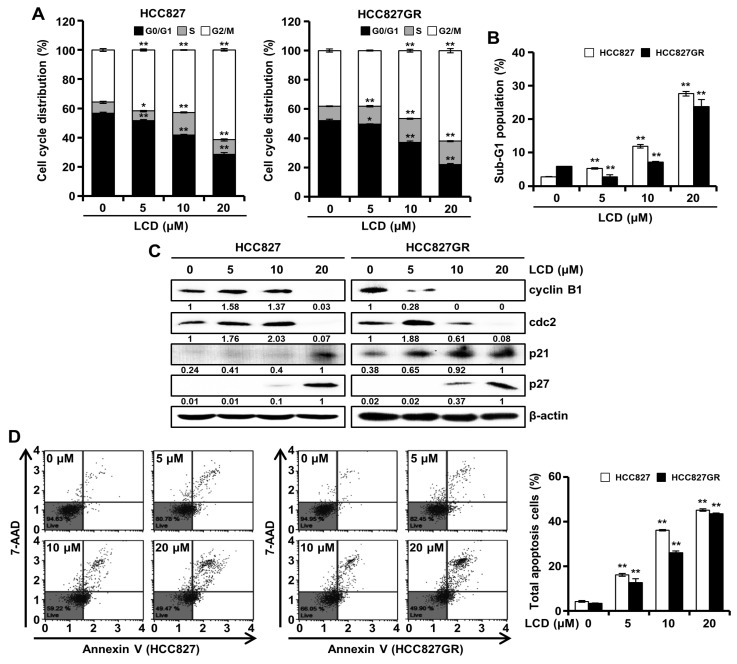
Cell cycle regulation and apoptosis induction of LCD. HCC827 and HCC827GR cells were exposed to LCD for 48 h. (**A**,**B**) Flow cytometry assay. LCD induced the arrest of non-small cell lung cancer (NSCLC) cells at the G2/M phase of cell cycle. In the G0/G1, S, G2/M, and sub-G1 phases, each value represents the mean ± SD (n = 3). (**C**) Expression levels of cell cycle-related proteins in HCC827 and HCC827GR cells were determined by Western blot. Relative protein levels of cyclin B1, cdc2, p21 and p27 were quantified using Image J software. (**D**) Apoptosis was determined using Annexin V/7-Aminoactinomycin D (7-ADD) staining. Annexin V/7-AAD double-stained cells were detected with a Muse™ Cell Analyzer. United Annexin V/7-ADD reactivity allowed the classification of cells into four groups: early apoptotic cells [Annexin V (+) and 7-AAD (−)], late apoptotic or dead cells [Annexin V (+) and 7-AAD (+)], dead cells [Annexin V (−) and 7-AAD (+)], and live cells [Annexin V (−) and 7-AAD (−)]. Data are presented as the mean ± SD of three independent experiments. *, *p* < 0.05 and **, *p* < 0.01.

**Figure 5 biomolecules-10-00297-f005:**
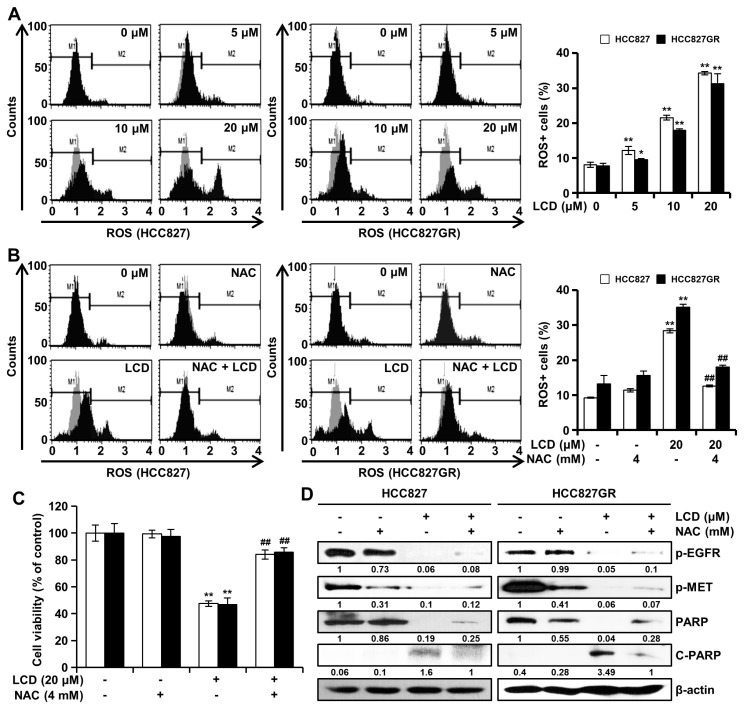
LCD treatment leads to intracellular reactive oxygen species (ROS) generation. (**A**) HCC827 and HCC827GR cells were treated with LCD (5, 10, and 20 μM) for 48 h, and intracellular ROS levels were determined using a Muse™ Cell Analyzer. (**B**) NSCLC cells were pretreated with 4 mM of N-acetylcysteine (NAC; an ROS inhibitor) for 3 h and then treated with 20 μM of LCD for 48 h. Values are expressed as means ± SD (n = 3). (**C**) MTT assay was performed to detect the viability of NSCLC cells treated with increasing concentrations of LCD and NAC. Values shown in the graph represent mean ± SD (n = 6). (**D**) After NAC pretreatment and LCD treatment, whole cell lysates were then subjected to Western blotting with antibodies against phosphorylated (p)-EGFR, p-MET, poly (ADP-ribose) polymerase (PARP), and β-actin. The intensity of the p-EGFR, p-MET, PARP, and C-PARP bands was quantified using Image J. *, *p* < 0.05 and **, *p* < 0.01 compared with the control. ##, *p* < 0.01 significantly different from LCD-treated cells.

**Figure 6 biomolecules-10-00297-f006:**
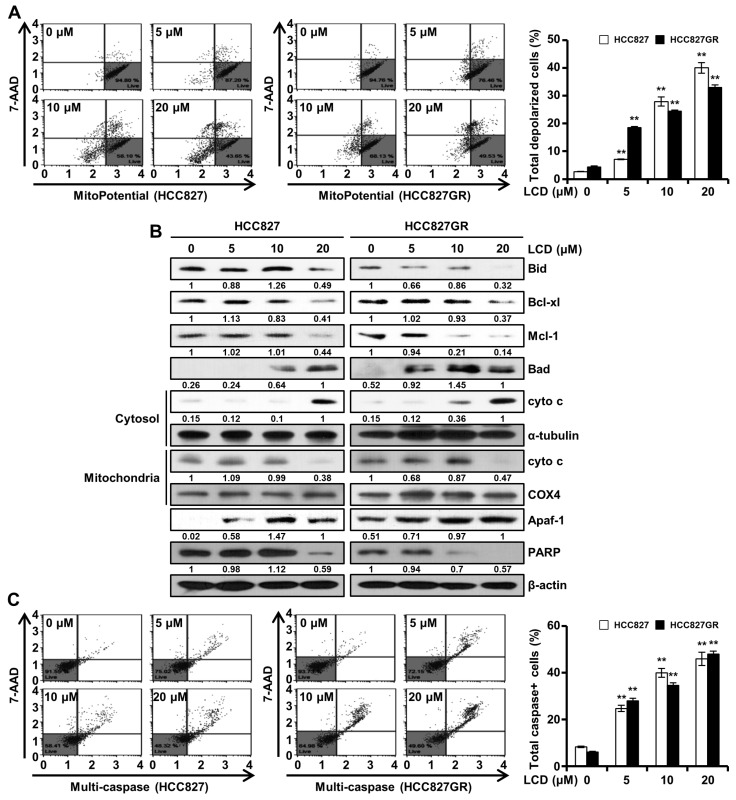
Effect of LCD on mitochondrial membrane potential (MMP) and activities of caspases in NSCLC cells. (**A**) Cells were exposed to LCD (5, 10, and 20 M) or DMSO for 48 h and MMP was analyzed by using a Muse™ Cell Analyzer. The fluorescence from right to left means the depolarization of MMP. Data are expressed as means ± SD of three independent experiments performed in triplicate. (**B**) After treatment of LCD, both cell lysates were subjected to Western blotting. Expression levels of Bid, Bcl-xl, Mcl-1, Bad, cyto c, α-tubulin, COX4, Apaf-1, and PARP were normalized to that of β-actin. Band intensities were quantified using Image J software. (**C**) HCC827 and HCC827GR cells were treated with an increasing concentration of LCD for 48 h. Multi-caspase (caspase-1, -3, -4, -5, -6, -7, -8, and -9) activity was determined using a Muse™ Cell Analyzer. Values are means (caspase-positive) ± SD. **, *p* < 0.01.

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
