# Peer review of "Licochalcone D Induces ROS-Dependent Apoptosis in Gefitinib-Sensitive or Resistant Lung Cancer Cells by Targeting EGFR and MET"

_biomolecules, 2020, doi:10.3390/biom10020297_

Round 1
Reviewer 1 Report
The article is of great interest and is well researched. The methods section is ample and well organized with regard to content and detail.
Flavonoids are, as you stated, important compounds that possess various properties, from antimicrobial to antioxidant. However, it is well established that, in general, their bioavailability in the human body is low.
For this reason, a very important question would be: Potentially, how well would licochalcone be absorbed in the human body?
It would also be interesting to know if various other phytochemical compounds contained in Chinese licorice, aside from flavonoids (such as saponins, for example) could synergize its proapoptotic and anticancer effects, especially in vivo.
Licochalcone presents us with an important phytochemical that should be included in future double-blinded and placebo-controlled trials with cancer patients, in order to ascertain its potential efficacy and safety in human participants.
Author Response
Thank you for suggestion. We've added your comment to the discussion part.
→ Line 448: In addition, while common flavonoids generally have low bioavailability in humans, LCs have been shown to be well absorbed through passive diffusion through the Caco-2 cell monolayer, a human intestinal cell line [26]. Because intestinal permeability is an important factor influencing the bioavailability of drugs, LCs can be expected to have high bioavailability in the human body. Liquiritin, another component of licorice, enhanced the cell proliferation inhibitory effect of cisplatin in gastric cancer cells resistant to cisplatin, and showed synergistic effects on tumor growth inhibition in vivo [27]. In the future, we will evaluate the anti-tumor efficacy of lung cancer cells with different resistance mechanisms for the combination of anti-cancer drugs and LCD.

Reviewer 2 Report
Shim et al. in their recent manuscript entitled: "Licochalcone D induces ROS-dependent apoptosis in gefitinib-sensitive or -resistant lung cancer cells by targeting EGFR and MET" describe the comprehensive analyses of LCD influence on lung cancer cell lines. In most cases, studies performed using one compound on a limited number of cell lines are of limited value. However, in this case, the studies are well designed and provide concise information about the compound's potential in vitro.
I have only several minor comments:
H20, CO2 (many typos) -> H2O, CO2 line 114: "each lane ... was loaded with proteins" - quite obvious, the question is if the amount of proteins loaded in each lane was equal and if yes, what was the amount 1.95x105 (as the example of many typos) -> 1.95×105 line 219-220 the information that ANOVA was followed by the Prism 5.0 (you mean GraphPad Prism?) package is insufficient. The caption of each graph that contains statistically analyzed data should contain clear information about what post-test was used. line 275: no SD for IC50 the authors represent only one Western blot analysis for each experiment. If these experiments were performed only once. If yes - clear information about that should appear in the manuscript, if no - the authors should include graphs densitometry, SD and statistical analysis In the pull-down assay, the authors checked the obtained samples for EGFR and MET. If any other proteins were present in the samples treated with LCD-beads. I would expect that membranes will be stained e.g. with Ponceau S to see if other proteins are present.Generally, the manuscript is well written, easy to understand the workflow and the idea of the studies. Since the manuscript does not contain any anticancer studies (in vivo) I would rather drop the conclusions about compound's "potential as an effective anti-cancer compound". It is definitely to early to make some assumptions/conclusions.
Author Response
Thank you for suggestion. We modified each point as advised.
Line 101: humidified atmosphere of 5% CO2. → humidified atmosphere of 5% CO2.
Line 114: Each lane in 8-15% SDS-PAGE gels was loaded with proteins. → The equal amount of protein was loaded on 8-15% SDS-PAGE gels.
Line 130: 7.5), 20 mM MgCl2, 0.1 mg/ml BSA, and 50 μM dithiothreitol (DTT), 2 mM MnCl2 and 100 μM sodium → 7.5), 20 mM MgCl2, 0.1 mg/ml BSA, and 50 μM dithiothreitol (DTT), 2 mM MnCl2 and 100 μM sodium
Line 174-177: To evaluate HCC827 (1.95x105) and HCC827GR (1.8x105) cell death with LCD treatment, Annexin V/7-AAD staining was performed using a Muse™ Annexin V & Dead Cell kit (MCH100105, Merck Millipore, Billerica, MA, USA). These cells were seeded onto a 6-well plate and treated with DMSO or LCD at different concentrations for 48 h. → To evaluate NSCLC cell death with LCD treatment, Annexin V/7-AAD staining was performed using a Muse™ Annexin V & Dead Cell kit (MCH100105, Merck Millipore, Billerica, MA, USA). The HCC827 (1.95x105) and HCC827GR (1.8x105) cells were seeded onto a 6-well plate and treated with DMSO or LCD at different concentrations for 48 h.
Line 183: Briefly, HCC827 (1.95x105 cells/well) and HCC827GR (1.8x105 cells/well) were collected by → Briefly, HCC827 and HCC827GR cells were collected by
Line 203: 10 mM HEPES (pH 8.0), 10 mM KCl, 1.5 mM MgCl2∙6H2O, 1 mM EDTA, 1 mM EGTA, 0.1 mM → 10 mM HEPES (pH 8.0), 10 mM KCl, 1.5 mM MgCl2∙6H2O, 1 mM EDTA, 1 mM EGTA, 0.1 mM
Prism 5.0 means GraphPad Prism. Added information about post-test in statistical analysis data.
Line 218-219: Statistical significance was evaluated using one-way analysis of variance (ANOVA) followed by the Prism 5.0 statistical package. → Statistical significance was evaluated using the software GraphPad Prism statistics (v5, GraphPad Software, USA, RRID: SCR_002798). Differences among multiple groups were tested using one-way or two-way ANOVA followed by Dunnett’s post hoc test.
The presence of other proteins in the samples treated with LCD-bead was evaluated. We confirmed the effect of LCD on AKT, the major signaling molecule of EGFR or c-MET, by pull-down analysis. There was no interaction between AKT and LCD. The data is added to figure 1B, and the result is written in line 227.
→ Line 228-229: However, there was no interaction between LCD and AKT.
We added the SD value of IC50 to the result lies 280-281 of the manuscript.
Line 280-281: IC50 values of LCD for viability of HCC827 and HCC827GR cells were 17.9 and 19.1 μM, respectively. → IC50 values of LCD for viability of HCC827 and HCC827GR cells were 17.9 ± 0.97 μM and 19.1 ± 0.5 μM, respectively.
Western blot was performed three times, showing the same pattern, and representative photographs. Protein quantitation was indicated by performing Image J software. We added content to the figure legend. We've also changed the figure to include the western blot.
→ Line 276: Band intensities were quantified using Image J.
→ Line 316-317: Relative protein levels of cyclin B1, cdc2, p21 and p27 were quantified using Image J software.
→ Line 348-349: The intensity of p-EGFR, p-MET, PARP and C-PARP bands was quantified using Image J.
→ Line 377: Band intensities were quantified using Image J software.
